# Gamma-Aminobutyric Acid Supplementation Boosts the Phytohormonal Profile in ‘*Candidatus* Liberibacter asiaticus’-Infected Citrus

**DOI:** 10.3390/plants12203647

**Published:** 2023-10-22

**Authors:** Yasser Nehela, Nabil Killiny

**Affiliations:** 1Department of Plant Pathology, Citrus Research and Education Center, University of Florida, 700 Experiment Station Rd., Lake Alfred, FL 33850, USA; yasser.nehela@agr.tanta.edu.eg; 2Department of Agricultural Botany, Faculty of Agriculture, Tanta University, Tanta 31527, Egypt

**Keywords:** huanglongbing, vector-borne diseases, emerging pathogens, GABA, salicylic acid (SA), auxin, jasmonic acid (*t*JA), abscisic acid (ABA)

## Abstract

The devastating citrus disease, Huanglongbing (HLB), is associated with ‘*Candidatus* Liberibacter sp.’ and transmitted by citrus psyllids. Unfortunately, HLB has no known sustainable cure yet. Herein, we proposed γ-aminobutyric acid (GABA) as a potential eco-friendly therapeutic solution to HLB. Herein, we used GC/MS-based targeted metabolomics combined with gene expression to investigate the role of GABA in citrus response against HLB and to better understand its relationship(s) with different phytohormones. GABA supplementation via root drench boosts the accumulation of endogenous GABA in the leaves of both healthy and ‘*Ca*. L. asiaticus’-infected trees. GABA accumulation benefits the activation of a multi-layered defensive system via modulating the phytohormone levels and regulating the expression of their biosynthesis genes and some pathogenesis-related proteins (*PRs*) in both healthy and ‘*Ca*. L. asiaticus’-infected plants. Moreover, our findings showed that GABA application stimulates auxin biosynthesis in ‘*Ca*. L. asiaticus’-infected plants via the activation of the indole-3-pyruvate (I3PA) pathway, not via the tryptamine (TAM)-dependent pathway, to enhance the growth of HLB-affected trees. Likewise, GABA accumulation was associated with the upregulation of SA biosynthesis genes, particularly the PAL-dependent route, resulting in higher SA levels that activated *CsPR1*, *CsPR2*, *CsPR5*, and *CsWRKY70*, which are prominent to activation of the SA-mediated pathway. Additionally, higher GABA levels were correlated with an enhanced JA profile and linked with both *CsPR3* and *CsPR4*, which activates the JA-mediated pathway. Collectively, our findings suggest that exogenous GABA application might be a promising alternative and eco-friendly strategy that helps citrus trees battle HLB.

## 1. Introduction

Citrus is one of the highest-value fruit crops in terms of international trade worldwide. However, citrus production is threatened by the vector-transmitted bacterial disease, Huanglongbing (HLB; also known as the citrus greening disease) [1,2,3]. Although Koch’s postulates have not been fulfilled yet, HLB was proposed to be associated with three phloem-limited, Gram-negative ‘*Candidatus* Liberibacter’ species, based on their 16S rDNA sequence and geographical distribution [2,4,5,6]. ‘*Ca.* L. asiaticus’ in Asia and the Americas [2,6] and ‘*Ca.* L. americanus’ in Brazil [1] are transmitted by the Asian citrus psyllid, *Diaphorina citri* Kuwayama (Hemiptera: Liviidae), whereas ‘*Ca.* L. africanus’ in Africa [7] is transmitted by the African psyllid, *Trioza erytreae* Del Guercio (Hemiptera: Triozidae) [2,6,7].

Because HLB is an incurable plant disease and has no known sustainable cure yet, its management strategies are mainly centered on diminishing the vector population within citrus groves or preventing the acquisition and transmission of ‘*Ca.* Liberibacter spp.’ using numerous tactics including, but not limited to, chemical control and insecticides [8,9,10], biocontrol using entomopathogenic agents such as the ascomycetous fungus *Isaria fumosorosea* [11,12,13,14,15], RNA interference (RNAi) [16,17,18,19,20], gene-targeting and editing using clustered regularly interspaced palindromic repeats, (CRISPR) and CRISPR-associated (Cas) proteins, particularly the CRISPR/Cas9 system [21,22], and recently using the antisense oligonucleotides 2′-deoxy-2′-fluoro-arabinonucleotide (FANA) RNA-targeting technology [23,24].

Nevertheless, ‘*Ca.* Liberibacter spp.’ cannot be targeted directly using the new control strategies (RNAi, FANA, and CRISPR/Cas9) because of its characteristic cellular immune lifestyle, life cycle, and replication systems [25]. Moreover, chemical- and antibiotic-based treatments of ‘*Ca.* Liberibacter spp.’ are challenging because the bacterial pathogen can produce protective biofilms, develop resistance to antimicrobial compounds, and make it difficult to reach the vascular tissue [26,27]. Thus, the development and implementation of sustainable eco-friendly alternative solutions to stop HLB is a necessity. Enhancing plant resilience through the fine-tuning of metabolic-based response(s), the maintenance of phytohormonal homeostasis, and the regulation of plant growth performance is a promising alternative solution that leads to plant stress tolerance.

Our previous studies showed that citrus plants retain several metabolic-based defense responses against ‘*Ca*. L. asiaticus’ and its insect vector [3,28,29,30]. These metabolic-based defense responses include, but are not limited to, leaf volatiles [31], amino, organic, and fatty acids [32], photosynthetic pigments [33], phytohormones [34,35], polyamines [36,37,38], phytomelatonin [39,40], the tricarboxylic acid- (TCA-) associated compounds, and non-proteinogenic amino acids such as γ-aminobutyric acid (GABA) [36].

The four-carbon non-proteinogenic amino acid, GABA, is ubiquitously distributed in all of the plant species analyzed so far [41]. In land plants, GABA is mainly biosynthesized in the cytosol from glutamate via a closed-loop short pathway known as the ‘GABA shunt’ that bypasses two steps from Ketoglutarate-to-succinate conversion outside the TCA cycle [36,42,43,44,45,46,47]. However, it could be synthesized through a non-enzymatic reaction from proline under oxidative stress [48] or via polyamine catabolism [36,49]. For example, GABA could be synthesized from putrescine and spermidine under specific stress conditions using D-amino oxidase (*DAO*) and polyamine oxidase (*PAO*), respectively [50]. In the GABA shunt, GABA is primarily synthesized from glutamate via an irreversible enzymatic reaction of cytosolic glutamate decarboxylase (*GAD*) with pyridoxal phosphate as a cofactor [51]. *GAD* catalyzes the decarboxylation of glutamic acid to form GABA. GAD binds to the Ca^2+^-dependent binding protein, calmodulin, which is regulated by the cytosolic Ca^2+^ levels and neutral pH in intact plants [51,52,53].

Moreover, the GABA shunt is functionally linked with the TCA cycle [36,45] via three evolutionarily conserved enzymatic steps. Firstly, the mitochondrial GABA permease (*gabP*) mediates the translocation of GABA from the cytosol into the mitochondria [36,48] where it is subsequently catabolized to succinic semialdehyde via the activity of GABA transaminase (*gabT*). Finally, succinic semialdehyde is catabolized to succinate, a key intermediate in the TCA cycle, by the activity of succinate semialdehyde dehydrogenase (*SSADH*; also known as *gabD*) [48]. It is worth mentioning that considerable homologies of *gabP*, *gabT*, and *gabD* have been frequently reported in both prokaryotes and eukaryotes [36,45,54].

GABA plays a multilayered function in plants. It has been proposed, with strong evidence, that GABA is not only a key regulator of primary and secondary metabolic pathways, particularly the TCA cycle and carbon/nitrogen metabolism, but also it acts as a key signaling molecule in plant growth and development [44,55,56]. For instance, GABA is required for pollen tube growth and guidance [57], as well as cell elongation [58]. Moreover, it has been reported that GABA contributes directly or indirectly to plant responses to various abiotic and environmental stresses such as salinity, drought, unfavorable temperature (heat and cold), and osmotic stress [59,60,61]. Furthermore, GABA plays a key role in plant defense against phytopathogenic microorganisms including fungi such as *Cladosporium fulvum* [62], *Magnaporthe grisea* [63], *Botrytis cinerea* [64,65], *Fusarium graminearum* [66,67], *Alternaria alternata* [68], *Rhizoctonia solani* [69], and *Lasiodiplodia theobromae* [70], as well as viruses such as Jatropha mosaic begomovirus [71] and Potato virus Y [72].

Likewise, GABA accumulation is associated with plant defense against several bacterial pathogens including Pseudomonas syringae [73,74], *Agrobacterium tumefaciens* [75], *Xylella fastidiosa* [76], and Ralstonia solanacearum [77]. In addition, GABA was reported to be involved in plant defense against invertebrate pests, particularly insect herbivores [53,78], such as *Choristoneura rosaceana* [79] and *Spodoptera littoralis* [80]. In agreement with the findings, using a targeted gas chromatography-mass spectrometry (GC-MS)-based method, we showed that GABA was accumulated to higher levels in ‘*Ca.* L. asiaticus’-infected and *D. citri*-infested citrus plants [32,36]. However, the molecular mechanisms behind the elevated GABA levels that help host plants cope with various phytopathogens are poorly understood.

In our previous study, we showed that exogenous GABA supplementation significantly enhanced the endogenous level of phytohormones in healthy *Citrus sinensis* [81]. Nevertheless, to the best of our knowledge, the physiological and biochemical roles of exogenous GABA application in HLB-affected citrus were not previously reported. Herein, we hypothesize that GABA supplementation might influence the phytohormonal homeostasis of the four phytohormones groups, including salicylates (benzoic acid [BA], *trans*-cinnamic acid [*t*CA], and salicylic acid [SA]), auxins (indole acetic acid [IAA], indole-3-butyric acid [IBA], indole propionic acid [IPA]), *trans*-jasmonic acid (*t*JA), and abscisic acid, as well as potentially regulating the transcript levels of their biosynthetic genes. Moreover, we suggest that exogenous GABA supplementation might have an antibacterial role that can diminish the ‘*Ca*. L. asiaticus’ population within the infected leaves. Collectively, these metabolic alterations will maximize the citrus’s ability to cope with the disease in HLB-affected areas and introduce the GABA application as a sustainable, eco-friendly, and cost-effective therapeutic solution against HLB disease.

## 2. Results

### 2.1. Exogenous GABA Application Boosted the Accumulation of Endogenous GABA in Both Healthy and ‘Ca. L. asiaticus’-Infected Plants

To confirm that exogenously applied GABA supports systemic GABA accumulation, we investigated the effect of GABA supplementation via root drench on the endogenous GABA levels in the leaves of both healthy and ‘*Ca*. L. asiaticus’-infected citrus plants (Figure 1). In general, exogenous GABA supplementation significantly enhanced the accumulation of endogenous GABA in both healthy and ‘*Ca*. L. asiaticus’-infected leaves (*p* < 0.0001 and *p* = 0.0191, respectively; Figure 1a,b). It is worth mentioning that although the presence of ‘*Ca*. L. asiaticus’ significantly increased the endogenous GABA levels in infected plants compared with healthy controls (*p* < 0.0001), and that GABA supplementation considerably boosted GABA accumulation in GABA-treated ‘*Ca*. L. asiaticus’-infected plants compared with GABA-treated healthy ones (*p* = 0.0109; Figure 1a,b).

### 2.2. GABA Accumulation Diminished the ‘Ca. L. asiaticus’ Population within the Infected Valencia Sweet Orange Leaves

The effect of GABA application (10 mM GABA) on the bacterial population of ‘*Ca*. L. asiaticus’ within the detached leaves of Valencia sweet orange was investigated using quantitative PCR (qPCR) and expressed as cycle threshold (C*_T_*) values, which negatively reflect the bacterial population within the infected tissues. Briefly, although GABA supplementation did not affect the C*_T_* values in healthy leaves, it significantly increased the C*_T_* values of treated infected leaves indicating a lower bacterial population of ‘*Ca*. L. asiaticus’ (Figure 1c).

### 2.3. GABA Supplementation Enhanced the Salicylate Content and Induced the Expression of Their Biosynthetic Genes in Both Healthy and ‘Ca. L. asiaticus’-Infected Plants

To better understand the physiological and biochemical roles of exogenous GABA application in HLB-affected citrus, the effect of GABA supplementation on three salicylate compounds including benzoic acid (BA, Figure 2A), *trans*-cinnamic acid (*t*CA, Figure 2B), and salicylic acid (SA, Figure 2C) was investigated. Generally, root drench application of 10 mM GABA stimulated the accumulation of BA (Figure 2A,B), *t*CA (Figure 2B,E), and SA (Figure 2C,F) in both healthy (*p* = 0.0035, *p* = 0.0027, and *p* = 0.0372, respectively) and ‘*Ca*. L. asiaticus’-infected plants (*p* < 0.0001, *p* = 0.0091, *p* = 0.0001, respectively). Interestingly, the ‘*Ca*. L. asiaticus’-infected trees treated with GABA had the highest BA, *t*CA, and SA levels throughout the experiment (2.1, 3.7, and 2.2-folds higher than the non-treated healthy plants, respectively).

Moreover, the effect of GABA supplementation on the transcript levels of 11 genes involved in SA biosynthesis (Figure 2G) in citrus leaves was investigated. Gene expression of all studied genes was normalized using four housekeeping genes including elongation factor-1 alpha (*CsEF-1α*), F-box/kelch-repeat protein (*CsF-box*), glyceraldehyde-3-phosphate dehydrogenase, cytosolic (*CsGAPC1*, aka *GAPDH*), and SAND family protein (*CsSAND*). The normalized gene expression using the four genes was very comparable to each other. Generally, the transcript levels of all studied SA biosynthesis-related genes were increased (up to 6.5-fold) after GABA supplementation with a greater effect on the ‘*Ca*. L. asiaticus’-infected plants (Figure 2H).

Two-way hierarchical cluster analysis (HCA) showed that all studied genes were clustered into three distinct clusters. “Cluster I” consists of four genes including chorismate synthase (*CsCS*), aspartate aminotransferase, cytoplasmic-like (*CsAST1*), phenylalanine ammonia-lyase (*CsPAL*), and chorismate mutase 2 (*CsCM2*) which were significantly higher in GABA-treated infected plants than other treatments. However, “Cluster II” consists of only three genes, including *CsAST2*, isochorismate synthase (*CsICS*), and alcohol acyltransferase (*CsAAT*), which were higher in non-treated and GABA-treated ‘*Ca*. L. asiaticus’-infected plants without significant differences between them (Figure 2H).

It is worth mentioning that the transcript levels of *CsPAL,* a key SA biosynthesis enzyme via the phenylpropanoid pathway, were induced in GABA-treated healthy (*p* = 0.0001) and infected plants (*p* = 0.0072); nevertheless, it was expressed at higher levels in the presence of ‘*Ca*. L. asiaticus’ (Figure 2I). On the other hand, the transcript level of *CsICS,* a key SA biosynthesis enzyme via the isochorismate pathway, was stimulated in GABA-treated healthy plants (*p* = 0.0015) but not ‘*Ca*. L. asiaticus’-infected ones and no significant differences were observed in the expression of *CsICS* between GABA-treated and non-treated ‘*Ca*. L. asiaticus’-infected plants (*p* = 0.6655, Figure 2J).

### 2.4. GABA Application Induced the Auxins Content and Induced the Expression of Their Biosynthetic Genes in Both Healthy and ‘Ca. L. asiaticus’-Infected Plants

The impact of exogenous GABA application on the endogenous levels of three auxins including indole acetic acid (IAA; Figure 3A), indole propionic acid (IPA; Figure 3B), and indole butyric acid (IBA; Figure 3C) were investigated. Similar to the salicylates profile, the root drench application of 10 mM GABA significantly increased the endogenous levels of IAA (Figure 3A,D), IPA (Figure 3B,E), and IBA (Figure 3C,F) in both healthy (*p* = 0.0073, *p* = 0.0108, and *p* = 0.0019, respectively) and *Ca*. L. asiaticus’-infected plants (*p* = 0.0298, *p* = 0.0160, and *p* = 0.0184, respectively). It is worth noting that the presence of ‘*Ca*. L. asiaticus’ facilitated the highest accumulation of IAA (2.2 folds), IPA (3.7 folds), and IBA (2.2 folds) in GABA-treated infected plants compared with non-treated healthy ones (Figure 3D, Figure 3E, and Figure 3F, respectively).

Moreover, the impact of exogenous GABA application on the transcript levels of 12-auxin biosynthesis-related genes (Figure 3G) in citrus leaves was investigated. The normalized gene expression using the four housekeeping genes (*CsEF-1α*, *CsF-box*, *CsGAPC1*, and *CsSAND*) were very similar to each other. The transcript levels of all the studied auxin biosynthesis-related genes were increased (up to 6.5-fold) after GABA supplementation with a greater effect in the ‘*Ca*. L. asiaticus’-infected plants (Figure 3H). Additionally, the total HCA-dendrogram showed that all studied genes were clustered into two major clusters (Figure 3H). “Cluster I” consist of only four genes, including anthranilate synthase alpha subunit 1 (*CsASA*), tryptophan aminotransferase-related protein 4-like (*CsTAA4*), indole-3-pyruvate monooxygenase YUCCA8 (*CsYUC8*), and indole-3-acetaldehyde oxidase-like (*CsAO1*; aka Acetaldehyde oxidase). Cluster II consists of eight genes, including anthranilate synthase beta subunit 2 (*CsASB*), tryptophan synthase-like (*CsTS*), tryptophan synthase alpha chain, chloroplastic-like (*CsTSA*), tryptophan synthase beta chain 1, chloroplastic-like (*CsTSB*), indole-3-pyruvate monooxygenase YUCCA2 (*CsYUC2*), tryptophan aminotransferase-related protein 2-like (*CsTAA2*), bifunctional nitrilase/nitrile hydratase NIT4A-like (*CsNIT4*), aromatic-L-amino-acid decarboxylase-like (*CsTDC1*; aka tryptophan decarboxylase) (Figure 3H).

The four genes of “Cluster I” were the *focus of interest.* Briefly, the transcript level of *CsASA*, an early enzyme catalyzing the first reaction in the tryptophan biosynthesis pathway, was significantly increased in GABA-treated healthy plants (*p* = 0.0001) but not in treated ‘*Ca*. L. asiaticus’-infected ones (*p* = 0.5376) (Figure 3I). Likewise, *CsAO1*, a key IAA biosynthesis enzyme via the tryptamine (TAM) pathway, was upregulated in GABA-treated healthy plants (*p* = 0.0008) but not in treated ‘*Ca*. L. asiaticus’-infected ones (*p* = 0.6068) (Figure 3J). On the other hand, both *CsTAA4* and *CsYUC8*, two key enzymes in the main IAA biosynthesis pathway via indole-3-pyruvate (I3PA), were highly expressed in both healthy (*p* = 0.0022 and *p* = 0.0001, respectively) and ‘*Ca*. L. asiaticus’-infected citrus plants (*p* = 0.00112 and *p* = 0.0017, respectively) when treated with 10 mM GABA (Figure 3K and Figure 3L, respectively).

### 2.5. Exogenous GABA Application Increased the Abscisic Acid Content and Upregulated the Expression of Its Biosynthesis-Related Genes in Healthy and ‘Ca. L. asiaticus’-Infected Plants

The impact of supplementary GABA on the endogenous levels of abscisic acid (ABA) was investigated (Figure 4A). Briefly, the root drench application of 10 mM GABA significantly increased the endogenous ABA content in both healthy (*p* = 0.0021) and ‘*Ca*. L. asiaticus’-infected plants (*p* = 0.0419) (Figure 4B). Interestingly, the presence of ‘*Ca*. L. asiaticus’ considerably enhanced the endogenous ABA levels in non-treated (*p* = 0.0059) and GABA-treated infected plants (*p* = 0.0136) compared with non-treated (control) and GABA-treated healthy, respectively (Figure 4B). Moreover, the ABA biosynthesis pathway (Figure 4C) was dissected. Generally, all studied ABA biosynthesis genes were upregulated after GABA supplementation with greater effect for ‘*Ca*. L. asiaticus’-infected plants (Figure 4D). The total HCA-dendrogram showed that the six ABA biosynthesis-related genes were split into two distinct clusters. Cluster-I includes zeaxanthin epoxidase (*CsZEP*), violaxanthin de-epoxidase (*CsVDE*) and neoxanthin synthase (*CsNSY*), whereas cluster-II includes 9-cis-epoxycarotenoid dioxygenase (*CsNCED*), short-chain alcohol dehydrogenase (*CsABA2*; aka xanthoxin dehydrogenase), and abscisic aldehyde oxidase (*CsAAO3*) (Figure 4D).

### 2.6. GABA Supplementation Induced the Accumulation of Trans-Jasmonic Acid and Upregulated the Expression of Its Biosynthesis Genes in Healthy and ‘Ca. L. asiaticus’-Infected Plants

The effect of GABA supplementation on the endogenous levels of *trans*-jasmonic acid (*t*JA) was investigated (Figure 5A). Briefly, exogenous GABA supplementation considerably enhanced the endogenous *t*JA levels in healthy (*p* = 0.0009) and ‘*Ca*. L. asiaticus’-infected plants (*p* < 0.0001, Figure 5A,B). Treatment of ‘*Ca*. L. asiaticus’-infected trees with 10 mm GABA via root drench significantly increased the endogenous *t*JA levels compared with non-treated (3.1-folds; *p* < 0.0001) and GABA-treated healthy controls (1.9-folds; *p* < 0.0001) (Figure 5B).

Additionally, the expression levels of nine *t*JA biosynthesis-related genes (Figure 5C) were investigated. All *t*JA biosynthesis-related genes were highly expressed (up to 6.6-fold) in both healthy and ‘*Ca*. L. asiaticus’-infected plants after the treatment with 10 mM GABA with greater effect in ‘*Ca*. L. asiaticus’-infected plants (Figure 5D). The total HCA-dendrogram showed that the nine *t*JA biosynthesis-related genes were separated into two distinct clusters, in addition to acyl-coenzyme A1 (*CsACX1*), which clustered alone in the bottom of the HCA-dendrogram. Cluster-I (C-I) consists of only two genes including ω-3 fatty acid desaturase (*CsFAD*) and enoyl-CoA hydratase, mitochondrial-like (*CsAIM1*), whereas Cluster-II consists of six genes including lipoxygenase (*CsLOX*), 3-ketoacyl-CoA thiolase, peroxisomal-like (*CsKAT*), allene oxide cyclase (*CsAOC*), 12-oxophytodienoate reductase 3 (*CsOPR3*), allene oxide synthase (*CsAOS*), and acetate/butyrate-CoA ligase AAE7 (*CsAAE7*), which were upregulated in ‘*Ca*. L. asiaticus’-infected trees were treated with 10 mm GABA than other treatments (Figure 5D).

### 2.7. Treatment with Exogenous GABA Altered the Transcript Levels of Pathogenesis-Related Proteins in Healthy and ‘Ca. L. asiaticus’-Infected Plants

To better understand the molecular mechanisms behind the defensive role of GABA in the treated plants, the transcript levels of eight pathogenesis-related proteins (PRs) were investigated, including pathogenesis-related protein PR-1 (*CsPR1*), *CsPR2*, *CsPR3*, *CsPR4*, *CsPR5*, *CsPR15*, pathogenesis-related protein STH-2-like (*CsSTH*-2), and activator of SA-dependent defense (*CsWRKY70*). It is worth mentioning that infection with ‘*Ca*. L. asiaticus’ significantly increased the expression levels of *CsPR1* (2.0-folds; *p* < 0.0001; Figure 6A), *CsPR2* (2.9-folds; *p* = 0.0055; Figure 6B), *CsPR3* (2.7-folds; *p* = 0.0033; Figure 6C), *CsPR4* (2.6-folds; *p* = 0.0007; Figure 6D), *CsPR5* (2.8-folds; *p* = 0.0006; Figure 6E), *CsPR15* (2.9-folds; *p* = 0.0047; Figure 6F), *CsSTH*-2 (3.1-folds; *p* = 0.0016; Figure 6G), and *CsWRKY70* (3.6-folds; *p* = 0.0018; Figure 6H) compared with the healthy control. Moreover, exogenous GABA application significantly enhanced the expression levels of *CsPR3* and *CsPR5* in both healthy (*p* = 0.0144 and *p* = 0.0153, respectively) and ‘*Ca*. L. asiaticus’-infected leaves (*p* = 0.0303 and *p* = 0.0230, respectively) (Figure 6C and Figure 6E, respectively). However, GABA supplementation did not alter the expression of both *CsPR1* and *CsPR4* in healthy citrus leaves (*p* = 0.1934 and *p* = 0.1272, respectively), whereas it did significantly in ‘*Ca*. L. asiaticus’-infected leaves (*p* = 0.0349 and *p* = 0.0143, respectively) (Figure 6A and Figure 6D, respectively). On the other hand, exogenous GABA application had no significant effect on the transcript levels of *CsPR2, CsPR15,* and *CsSTH*-2 in either healthy or control plants (Figure 6B, Figure 6F and Figure 6G, respectively).

### 2.8. Exogenous GABA Supplementation Heightened the Multiple Correlations between Endogenous GABA and Phytohormones in Both Healthy and ‘Ca. L. asiaticus’-Infected Plants

To better understand the relationship between endogenous GABA and studied phytohormones, and to measure the strength of this relationship, multiple correlation analysis (MCA) was carried out (Figure 7). In healthy plants, and based on the correlation coefficient (*r*), MCA showed that exogenous GABA supplementation highly strengthened the correlation between endogenous GABA (as an independent variable) and SA (*r* = 0.91 compared with 0.29 in non-treated controls) and IAA (*r* = 0.85 compared with 0.15 in non-treated controls) (Figure 7A). Likewise, GABA supplementation greatly strengthened the positive correlation between endogenous GABA and BA (*r* = 0.75 compared with 0.38 in non-treated ‘*Ca*. L. asiaticus’-infected), SA (*r* = 0.96 compared with 0.62 in non-treated ‘*Ca*. L. asiaticus’-infected), and IAA (*r* = 0.75 compared with 0.10 in non-treated ‘*Ca*. L. asiaticus’-infected) (Figure 7B).

## 3. Discussion

The non-proteinogenic amino acid, GABA was reported in plants more than seven decades ago when Steward and his colleagues initially reported it as a major nitrogenous compound in the potato tubers [82]. Since this time, the physiological roles of GABA as a primary metabolite in plant growth and development, as well as stress responses have been extensively studied [44,55,56,57,58,59,60,61]. GABA regulates numerous primary and secondary metabolic pathways, particularly the TCA cycle, and acts as a signaling molecule in plant growth and development [44,55,56]. Moreover, GABA accumulates in response to various abiotic stress [59,60,61], as well as biotic stressors such as insect herbivores [53,78,79,80], viral diseases [71,72], fungal diseases [62,63,64,65,66,68,69,70], and bacterial pathogens [32,36,73,74,75,76,77]. The defensive role of GABA against phytopathogens might be due to its direct inhibitory activity, the induction of downstream defense responses, or a combination of both mechanisms. However, the biochemical and molecular mechanisms behind how the elevated GABA levels help host plants cope with various phytopathogens are poorly understood and several aspects of the complexity of the GABA signaling system remain ambiguous.

Recently, the role of versatile GABA in plants was well-documented and reviewed [56]. However, the authors ended their review article with some open questions that are still unclear or controversial and require future research to better understand the role of GABA in plants. One of these questions was “How do plant hormones interact with GABA?” [56]. Our findings in this study lay a brick to answer this question. Our findings showed that GABA accumulation enhanced the endogenous levels of phytohormones including three salicylates (BA, *t*CA, and SA), three auxins (IAA, IPA, and IBA), *t*JA, and ABA, and induced the expression levels of their biosynthetic genes in both healthy and ‘*Ca*. L. asiaticus’-infected plants. Although very few reports are available about the role of GABA accumulation in hormone metabolism, particularly ethylene (ET) [58,83], to the best of our knowledge, the crosstalk between GABA and different phytohormone groups has never been reported previously in citrus plants challenged with *Ca*. L. asiaticus’ infection.

Moreover, multiple correlation analysis showed that exogenous GABA supplementation highly strengthened the correlation between GABA, SA, and IAA in both GABA-treated healthy and ‘*Ca*. L. asiaticus’-infected plants compared with non-treated controls. In agreement with these findings, β-aminobutyric acid (BABA), a common GABA isomer, was reported to induce plant defense responses in many different plant species against a wide range of abiotic and biotic via the modulation of ABA and other phytohormones [84,85]. Briefly, BABA boosts the primary metabolism via the TCA cycle, phenylpropanoids, and octadecanoic pathways, as well as some phytohormones and indolic compounds [84]. For instance, free SA and its conjugated forms, SA 2-O-beta-D-glucose (SAG) and SA glucose ester (*SGE*) were accumulated at 24- and 48 h post-treatment (hpt), and the induction was higher by BABA than the avirulent *Pseudomonas syringae* pv *tomato* (*PstAvrRpt2*) [84]. Likewise, the endogenous levels of IAA and other indolic compounds included indole-3-carboxaldehyde (I3CHO), indole-3-acetamide (IAM), indol-3-pyruvic acid (I3PA), indole-3-acetyl-L-Ala (IALA), and indole-3-carboxylic acid methyl ester (I3CAME), were increased at 24 hpt with BABA [84]. It is worth mentioning that although the GABA biosynthesis pathway is well-established in higher plants, BABA is considered a xenobiotic since there is no evidence for its metabolism in plants [85,86,87]. However, a recent study reported BABA as a natural non-proteinogenic amino acid that is found in trace amounts under normal growth conditions (just below 50 ng g^−1^) in leaves and roots from various plant species including *Arabidopsis thaliana*, Brassica rapa, *Zea mays*, *Triticum aestivum* and *Physcomytrella patens* [88]. However, further investigations are required to confirm the biosynthesis of BABA in plants.

It is widely accepted that SA is produced in plants in the shikimate–phenylpropanoid pathway via two distinct routes, both starting from the intermediate chorismate using isochorismate synthase (*ICS*) or phenylalanine ammonia-lyase (*PAL*) [89]. In the first route, SA is synthesized through two enzymatic reactions starting with the conversion of chorismate to isochorismate using *ICS* [90,91], and then to SA using isochorismate pyruvate lyase (*IPL*) [92]. Although *CsICS* is well characterized in citrus [93] and was upregulated in ‘*Ca*. L. asiaticus’-infected plants [34,35], our findings from the current study showed that GABA supplementation significantly increased the transcript level of *CsICS* in treated healthy plants, but not ‘*Ca*. L. asiaticus’-infected ones. Moreover, *IPL* has only been reported from bacteria [94] and there is no evidence of its presence in citrus. Acknowledging the fact that the *ICS*-based SA biosynthesis pathway is relatively new in plants, and it is better established in bacteria [89], the PAL-dependent route was thought to be the common SA biosynthesis pathway.

In the second route, SA is synthesized from the amino acid _L_-phenylalanine to CA using *PAL* [89,92] then to BA, and finally to SA using benzoic acid-2-hydroxylase (*BA2H*) [89,95,96]. In the current study, *CsPAL* was upregulated in both GABA-treated healthy and infected plants; however, it was expressed at higher levels in the presence of ‘*Ca*. L. asiaticus’. Likewise, the upregulation of *CsPAL* was reported to be associated with the accumulation of SA in ‘*Ca*. L. asiaticus’-infected and *D. citri*-infested plants [34,35]. Even though the soluble oxygenase, *BA2H*, was partially purified and characterized from *Nicotiana tabacum* and suggested to catalyze the SA biosynthesis [97], the gene encoding this enzyme is not cloned yet and there is no evidence for the presence of *BA2H* within the citrus genome. Collectively, although the absence of both *IPL* and *BA2H* confounds the SA biosynthesis, our findings suggest that the PAL-dependent pathway might be responsible for the biosynthesis of SA from _L_-phenylalanine via CA and BA. Interestingly, our findings from targeted metabolomics showed that GABA supplementation enhanced the endogenous content of both compounds (CA and BA) in healthy and ‘*Ca*. L. asiaticus’-infected plants.

Isotope-based biochemical studies on other plant species including sunflower, potato, tomato, and pea, suggest that SA is mainly produced from cinnamate via the PAL-dependent pathway [98,99,100]. For example, SA was found to be synthesized from carboxyl-labeled benzoic acid in sunflowers, potatoes, and peas [98]. Likewise, labeled SA has been detected in *Primula acaulis* and *Gaultheria procumbens* leaves after the treatment with ^14^C-labeled benzoate suggesting a PAL-dependent pathway as a key route for SA biosynthesis [100]. In agreement with these results, silencing of *PAL* genes in tobacco or chemical inhibition of PAL activity in Arabidopsis, cucumber, and potato suggest that cinnamate-derived SA via the PAL-dependent pathway is the most important route for SA biosynthesis during pathogen infection [92]. Taken together, these findings support our hypothesis that the PAL-dependent pathway is a favorable route for SA accumulation in ‘*Ca*. L. asiaticus’-infected plants after the treatment with exogenous GABA.

Moreover, *PAL* was suggested to be an upstream enzyme that advances several other defense-related molecules [101]. In the current study, the accumulation of SA and upregulation of *CsPAL* were associated with the upregulation of eight PRs including *CsPR1*, *CsPR2*, *CsPR3*, *CsPR4*, *CsPR5*, *CsPR15*, *CsSTH*-2, and activator of SA-dependent defense (*CsWRKY70*). PR proteins are the major molecules in the plant immune system, particularly systemic acquired resistance (SAR), and are commonly employed as molecular biomarkers of defense signaling pathways [102]. It is worth mentioning that PR1, PR2 & PR5 are well-reported to be increased locally, as well as systematically, paving the road to SAR, activated by non-expressor of pathogen-related protein 1 (NPR1) and associated with SA-mediated pathway, whereas PR3, PR4 & PR12 are produced only locally to lead to local acquired resistance (LAR) via JA-mediated pathway [103].

Interestingly, our findings showed that GABA supplementation induced the accumulation of *t*JA and upregulated the expression of its biosynthesis genes in healthy and ‘*Ca*. L. asiaticus’-infected plants. Moreover, our previous study showed that the endogenous levels of *t*JA and its precursor linolenic acid, as well as the transcript levels of JA biosynthesis genes, were increased in citrus leaves upon *D. citri*-infestation greater than ‘*Ca*. L. asiaticus’-infection [32,34]. JA is the key component of the JA/ET-mediated pathway (aka wound response pathway) which was previously proven to be associated with defense against necrotrophic phytopathogens and insect herbivory [104,105,106,107,108]. For example, exogenous application of JA and/or its derivative, methyl jasmonate (me-JA), enhanced the defense mechanisms of Arabidopsis plants against beet armyworm, *Spodoptera exigua* [109] and negatively affected the performance of pistachio psyllid *Agonoscena pesticide* [110]. Taken together, these findings suggest that GABA induction of the JA-mediated pathway might play a key role in citrus’s defensive response against insect herbivory such as *D. citri*, as well as necrotrophic pathogens.

Furthermore, our findings showed that GABA application induced the auxins content including IAA, IBA, and IPA, and induced the expression of their biosynthetic genes in both healthy and ‘*Ca*. L. asiaticus’-infected plants. Gene expression results suggest that auxin biosynthesis in GABA-treated, ‘*Ca*. L. asiaticus’-infected plants might be mainly through the indol-3-pyruvic acid (I3PA) pathway using *CsTAA* and *CsYUC* since the transcription levels of both *CsTDC* and *CsAO1*, two main IAA biosynthesis enzymes in the tryptamine (TAM) pathway, did not change significantly in ‘*Ca*. L. asiaticus’-infected leaves after GABA application. Although auxins are well-known to be involved in plant growth and development, they also play a key role in plant response to biotic and abiotic stress [111]. Previously, we showed that the endogenous auxin levels and their precursor, tryptophan, were increased as a part of citrus response against infection with ‘*Ca*. L. asiaticus’ and/or infestation with *D. citri* [32,34]. Auxin’s accumulation in ‘*Ca*. L. asiaticus’-infected and/or *D. citri*-infested leaves were supported by the upregulation of auxins biosynthetic genes [34]. Likewise, IAA was significantly accumulated in ‘*Ca*. L. asiaticus’-infected fruits [112,113] and flowers [114]. Moreover, IAA was reported to be associated with plant defense against insect herbivory via interaction with JA [115]. JA enhances auxin biosynthesis and vice versa [115]. Together, these findings suggest that in addition to the main role of auxins in citrus growth and development, they might be involved with other phytohormone-based defensive responses against ‘*Ca*. L. asiaticus’ and its insect vector.

Additionally, our findings showed that exogenous GABA application increased the ABA content and upregulated the expression of its biosynthesis-related genes in healthy and ‘*Ca*. L. asiaticus’-infected plants. ABA is mainly involved in plant tolerance to abiotic stress such as drought, salinity, and extreme temperature, as well as plant responses against biotrophic pathogens, necrotrophic pathogens, and insect herbivory [104,107]. Our previous study showed that ‘*Ca*. L. asiaticus’ infection significantly induced the accumulation of ABA and its precursor zeaxanthin [33,34]. Similarly, infection with the ascomycetous fungus, *Colletotrichum acutatum*, significantly increased the endogenous ABA levels in citrus fruits and petals [114]. Collectively, we hypothesize that GABA supplementation might enhance citrus response against ‘*Ca*. L. asiaticus’ and other biotic and abiotic stressors.

The crosstalk between different phytohormones (SA, JA, ABA, and auxins) is controversial. For instance, ABA was reported as a negative regulator of both SA- and JA/ET-mediated pathways [104,107]. On the other hand, ABA positively regulated JA against herbivores but negatively affected SA against certain phytopathogens [116]. Previously, we proved that auxins and ABA accumulations were accompanied by higher levels of SA and JA in ‘*Ca*. L. asiaticus’-infected and *D. citri*-infested leaves, respectively. In the current study, GABA supplementation significantly enhanced endogenous SA, JA, ABA, and auxin levels. Enhanced phytohormone profile might be for the benefit of citrus response against multiple biotic attackers and environmental stressors as well.

To summarize our findings, a hypothetical model for the potential effects of GABA supplementation on different phytohormonal pathways and their roles in citrus defense responses is proposed and presented in Figure 8. Briefly, exogenous GABA application via root drench and ‘*Ca*. L. asiaticus’-infection enhances the accumulation of endogenous GABA in citrus leaves. Higher GABA levels benefit the development of a phytohormone-based multi-layered defensive system via promoting the endogenous content of stress-associated phytohormones, including the auxins SA, JA, and ABA, and their biosynthetic genes. Induced auxin levels in GABA-treated ‘*Ca*. L. asiaticus’-infected plants might be due to the activation of the I3PA pathway, not the TAM-dependent pathway. Higher auxin levels are involved in enhancing the growth of HLB-affected trees via several biochemical and physiological modifications, yet to be identified. Moreover, the upregulation of SA biosynthesis genes, particularly PAL-dependent route, resulted in higher SA levels that activate NPR1 and subsequently *CsPR1*, *CsPR2*, *CsPR5*, and *CsWRKY70* which contribute to the activation of SA-mediated pathway against biotrophic pathogens (Figure 8). On the other hand, enhanced JA levels could be associated with the activation of *CsPR3* and *CsPR4* activates the JA-mediated pathway against insect herbivory and necrotrophic phytopathogens. Finally, the application of GABA enhances the ABA biosynthesis pathway to protect citrus plants against abiotic stress (Figure 8).

## 4. Materials and Methods

### 4.1. Plant Materials and Growth Conditions

Throughout this study, the most commonly grown commonly grown HLB-susceptible cultivar, ‘Valencia’ sweet orange (*Citrus sinensis*), was utilized as an experimental plant. All trees were approximately 95 ± 5 cm in height and approximately 18 months old at the time of sampling. All plant materials were maintained in a secured greenhouse at the Citrus Research and Education Center, University of Florida (CREC-UF), Lake Alfred, Florida, at 27.5 ± 2.5 °C, with 65 ± 5% relative humidity, and 16:8 h of light/dark period. All plants were watered twice weekly and fertilized monthly with a water-soluble 20:10:20 NPK fertilizer (Allentown, PA, USA). ‘*Ca*. L. asiaticus’-infected plants were obtained via graft-inoculation of 12-month-old healthy plants with ‘*Ca*. L. asiaticus’-positive materials and maintained under the same conditions described above. About six months later, the presence of ‘*Ca*. L. asiaticus’ was confirmed using PCR according to Tatineni et al. [117] upon the development of the initial HLB symptom.

### 4.2. Treatment of Citrus Plants with Exogenous GABA and Leaf Sampling

To investigate the effect of GABA supplementation on the endogenous GABA levels, phytohormonal homeostasis, and phytohormones biosynthesis genes of citrus leaves, 18- months old Valencia trees (healthy versus ‘*Ca.* L. asiaticus’-infected) were treated with 300 mL of aqueous GABA solution (Sigma-Aldrich, St. Louis, MO, USA) per tree via soil drench method. The final concentration of GABA was adjusted to 10 mM based on our previous studies [81,118]. The solution that surpassed the soil field capacity, and gravitationally drained from the pot bottom, was collected using a plastic plate and reapplied to the plant. Distilled water was used as a negative control. All healthy and ‘*Ca.* L. asiaticus’-infected plants (GABA-treated vs. non-treated) were maintained under the same conditions described above for seven days. Our previous studies showed that endogenous GABA content, as well as the endogenous phytohormones levels, reached their highest peak at 7 days post-treatment (dpt) [81,118]. Thus, 7-dpt, three leaves were collected from each biological replicate, chopped, pooled together, and immediately kept at −80 °C until further analysis.

### 4.3. Effect of GABA Supplementation on ‘Ca. L. asiaticus’ Titer in Valencia Sweet Orange Detached Leaves

The antibacterial activity of GABA against ‘*Ca*. L. asiaticus’ was examined using the detached Valencia sweet orange leaves assay as described in our previous study [40]. Briefly, fully mature healthy or HLB-symptomatic leaves were detached with their complete petioles from 18-month-old Valencia sweet orange trees, covered immediately with wet absorbent gauze, placed on ice, and transferred to the laboratory. To investigate the effect of GABA supplementation on ‘*Ca*. L. asiaticus’ titer, the detached leaves were incubated separately in 10 mM GABA aqueous solution, in addition to the control (mock; Milli-Q water without GABA). All treatments were kept in the same growth conditions as described above. For sampling, a single leaf disc (~31 mm^2^) was collected from each leaf using a paper hole puncher before the treatment (zero time) and at 72 hpt. The bacterial titer of ‘*Ca*. L. asiaticus’ was quantified using a reverse transcription quantitative PCR (RT-qPCR) as described in our previous study [40].

### 4.4. Targeted Analysis of GABA and Phytohormones

Citrus leaf metabolites were extracted as described in our previous studies [32,34,35,36,119]. Briefly, approximately 100 ± 3 mg of each biological sample was ground in liquid nitrogen, then tertiarily extracted using 750 µL of acidic methanol 80%. Subsequently, the supernatants were collected, combined together, and then concentrated to 50 µL under a nitrogen stream. Right before derivatization, 5 µL of heptadecanoic acid (250 ppm) was added to each sample as an internal standard. All samples and standards were derivatized using methyl chloroformate (MCF) following the protocol of [120] with slight modifications as described in our previous studies [32,34,35,36,119].

For GC-MS analysis, 1 µL of the derivatized samples/standards was injected into a GC-MS system-model Clarus 680 (Perkin Elmer, Waltham, MA, USA) fitted with a ZB-5MS GC column (5% Phenyl-Arylene 95% Dimethylpolysiloxane; 30 m × 0.25 mm × 0.25 µm film thickness; Phenomenex, Torrance, CA, USA) and running in the selected ion monitoring (SIM) mode. Helium was used as the carrier gas at a suitable flow rate of 1 mL per minute. The GC thermo-program, MS ion identification, data acquisition, and chromatograms analysis were carried out as described in our previous studies [32,36]. Briefly, GABA and eight phytohormones (BA, *t*CA, SA, IAA, IBA, IPA, *t*JA, and ABA) were first identified by comparing their mass spectra with spectra from published literature [32,36]. The identification was verified using library entries of the ninth edition of the Wiley mass spectral database (John Wiley and Sons, Inc., Hoboken, NJ, USA), NIST 2011 mass spectral database (National Institute of Standards and Technology, Gaithersburg, MA, USA), and the Golm Metabolome Database (http://gmd.mpimp-golm.mpg.de/ accessed on 5 October 2020). Finally, the identification of targeted compounds was further confirmed by comparing their retention times (RT), linear retention indices (LRIs), and mass spectra with those of authentic reference standards (Sigma-Aldrich, St. Louis, MO, USA) treated identically to samples. Calibration curves, using a series of reference standards (0, 5, 10, 25 and 50 ppm) derivatized and treated identically to samples, were used to quantify targeted metabolites.

### 4.5. Gene Expression Analysis

The transcript levels of 47 genes (Appendix A) were determined in leaves of healthy and ‘*Ca*. L. asiaticus’-infected plants (GABA-treated vs. nontreated). These genes included 11 involved in SA biosynthesis, 12 genes involved in auxins biosynthesis, six genes involved in ABA biosynthesis, and nine genes involved in *t*JA biosynthesis. In addition, the transcript levels of eight pathogenesis-related proteins included *CsPR1*, *CsPR2*, *CsPR3*, *CsPR4*, *CsPR5*, *CsPR15*, *CsSTH-2*, and *CsWRKY70* (Appendix A), were investigated. Briefly, total RNA was extracted using TriZol^®^ reagent (Ambion^®^, Life Technologies, Grand Island, NY, USA) as described in our previous studies [32,36]. Then, a superScript first-strand synthesis system (Invitrogen) with random hexamer primers was used to synthesize cDNA as described by the manufacturer’s instructions. SYBR Green PCR master mix (Applied Biosystems, Foster City, CA, USA) was used to perform the qPCR on an ABI 7500 Fast-Time PCR System (Applied Biosystems, Foster City, CA, USA). Samples were analyzed in triplicate for each biological replicate for each treatment. The 2^−ΔΔ^*^C^*_T_ method was used to determine the relative changes in gene expression among PCR products [121]. Four housekeeping genes (reference genes) were used for the normalization of gene expression including elongation factor-1 alpha (*CsEF-1α*), F-box/kelch-repeat protein (*CsF-box*), glyceraldehyde-3-phosphate dehydrogenase, cytosolic (*CsGAPC1*, aka GAPDH), and SAND family protein (*CsSAND*) [122,123].

### 4.6. Statistical Analysis

Completely randomized design (CRD) was used as an experimental design in all experiments throughout this investigation. Four treatments included healthy (‘*Ca*. L. asiaticus’-free) versus ‘*Ca*. L. asiaticus’-infected citrus plants (GABA-treated vs. nontreated [control]) with six biological replicates for each treatment, were analyzed in duplicate (two technical replicates for each). The technical replicates were used only to test the reproducibility and variability of our extraction and derivatization protocols, as well as the reproducibility of the GC-MS instrument, but were not used for statistical analysis. A two-tailed *t*-test was used for pairwise statistical comparison between every two treatments including healthy versus ‘*Ca*. L. asiaticus’-infected, non-treated versus GABA-treated, and statistical significance was established as *p* < 0.05. Moreover, two-way hierarchical cluster analysis (HCA) associated with heatmaps was used to analyze the differences in transcript levels of various genes involved in the biosynthesis of different phytohormone groups. Distance and linkage of HCA were conducted using Ward’s minimum variance method [124]. Finally, multiple correlations between different phytohormones and GABA were calculated, and the correlation coefficients (*r*) were visualized as a circular graph of the correlation matrix.

## 5. Conclusions

GABA is a well-known signaling metabolite, nevertheless, the potential effect(s) of GABA on phytohormones in GABA-treated ‘*Ca*. L. asiaticus’-infected plants are limited, if not lacking. The findings of the current study explain the molecular and biochemical mechanisms of GABA-mediated response and prove its crosstalk with different phytohormone groups. To the best of our knowledge, the possible role of GABA against bacterial phytopathogens in general, and ‘*Ca.* L. asiaticus’ particularly, has not been reported previously. Exogenous GABA application might be a promising alternative eco-friendly strategy that helps citrus trees battle HLB particularly and other diseases in general. Ultimately, a better interpretation of the GABA-based defense mechanism(s) in citrus might draw a comprehensive understanding of phytohormones-based responses, which is critical for developing sustainable management strategies for HLB. However, further studies are required to study the best delivery method of GABA, as well as its effect on citrus horticultural response in both greenhouse and field evaluations. Moreover, future research will most likely be strengthened by deciphering the GABA-hormones signaling system in citrus.

## Figures and Tables

**Figure 1 plants-12-03647-f001:**
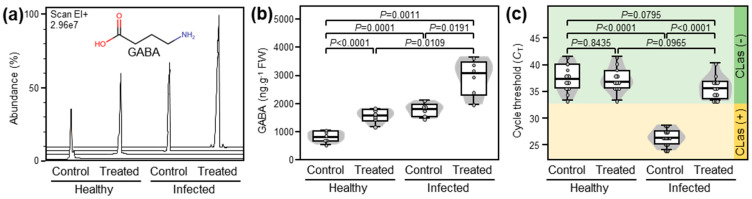
Effect of γ-aminobutyric acid (GABA) supplementation on the endogenous GABA levels and the bacterial titer in the leaves of healthy and ‘*Ca*. L. asiaticus’-infected Valencia sweet orange (*C. sinensis*) after derivatization with MCF using targeted GC-MS-SIM. (**a**) Representative chromatograms of GABA from healthy and *Ca*. L. asiaticus’-infected (infected) leaves without GABA treatment (Control) or after the treatment with 10 mM GABA (treated). (**b**) Endogenous GABA content (ng g^−1^ FW) from healthy and *Ca*. L. asiaticus’-infected (infected) leaves without GABA treatment (Control) or after the treatment with 10 mM GABA (treated). (**c**) Cycle threshold (C*_T_*) of reverse transcription quantitative PCR (RT-qPCR) for the detection of ‘*Ca*. L. asiaticus’ in HLB-symptomatic detached leaves of Valencia sweet orange after the treatment with 10 mM GABA (treated). The minimum and the maximum values are presented by whiskers, while thick horizontal lines specify the median. Boxes show the interquartile ranges (25th to 75th percentile of the data), white dots represent the raw data (*n* = 6), and gray shading represents the corresponding violin plot. Presented *p*-values are based on a two-tailed *t-*test to statistically compare each pair of treatments. Statistical significance was established as *p* < 0.05.

**Figure 2 plants-12-03647-f002:**
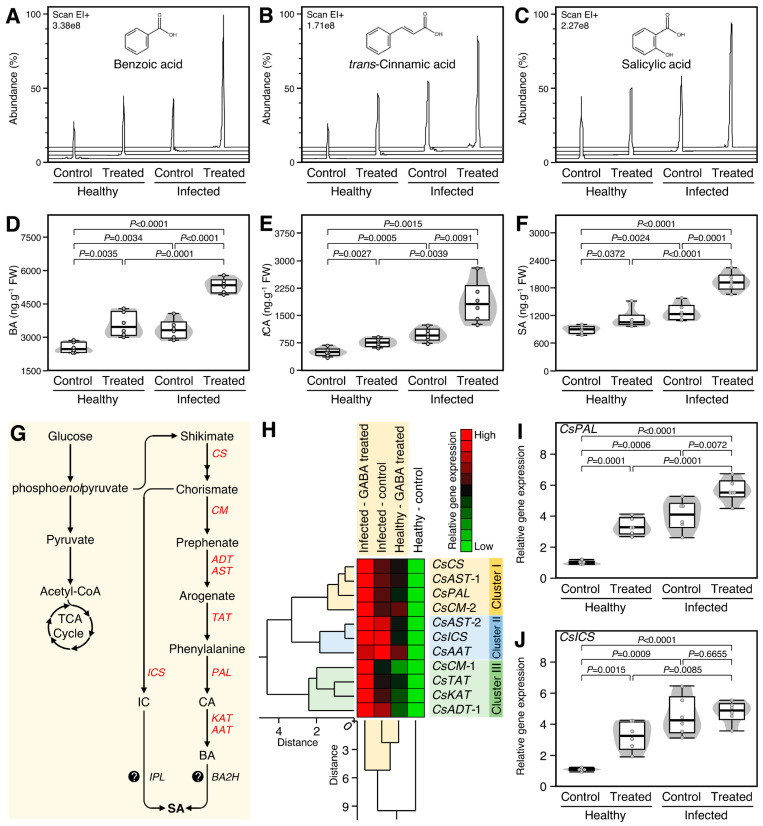
Effect of γ-aminobutyric acid (GABA) supplementation on the endogenous salicylates level and their biosynthesis genes in the leaves of healthy and ‘*Ca*. L. asiaticus’-infected Valencia sweet orange (*C. sinensis*). (**A**–**C**) Representative chromatograms of benzoic acid (BA), *trans*-cinnamic acid (*t*CA), and salicylic acid (SA), respectively, from healthy and *Ca*. L. asiaticus’-infected (infected) leaves without GABA treatment (Control) or after the treatment with 10 mM GABA (treated). (**D**–**F**) Endogenous contents (ng g^−1^ FW) of BA, *t*CA, and SA, respectively, from healthy and *Ca*. L. asiaticus’-infected (infected) leaves without GABA treatment (Control) or after the treatment with 10 mM GABA (treated). (**G**) Schematic representation of the SA biosynthesis pathway. (**H**) Relative gene expression of SA biosynthesis genes in healthy and *Ca*. L. asiaticus’-infected (infected) leaves without GABA treatment (Control) or after the treatment with 10 mM GABA (GABA treated). The transcript levels were organized using two-way hierarchical cluster analysis (HCA) using standardized gene expression patterns using Ward’s minimum variance method and presented as a heat map. Rows represent the SA biosynthetic genes, while columns represent different treatments. Low and high expression levels are indicated with green and red colors, respectively (see the scale at the top right corner). The full list of expressed genes, names, accession numbers, and primers is available in Appendix A. (**I**,**J**) Relative gene expression of phenylalanine ammonia-lyase (*CsPAL*) and isochorismate synthase (*CsICS*), respectively. The minimum and the maximum values are presented by whiskers, while thick horizontal lines specify the median. Boxes show the interquartile ranges (25th to 75th percentile of the data), white dots represent the raw data (*n* = 6), and gray shading represents the corresponding violin plot. Presented *p*-values are based on a two-tailed *t*-test to statistically compare each pair of treatments. Statistical significance was established as *p* < 0.05.

**Figure 3 plants-12-03647-f003:**
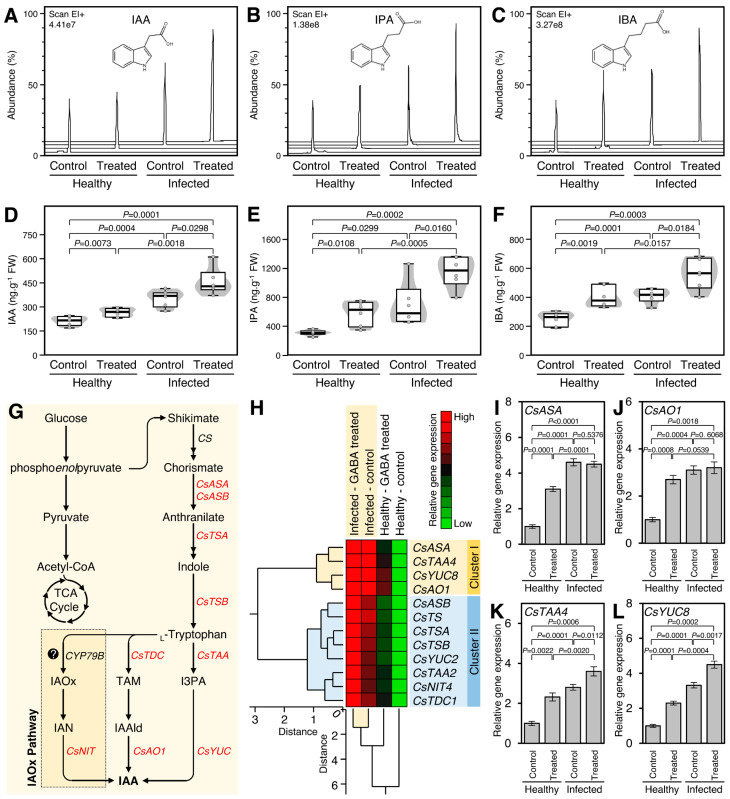
Effect of γ-aminobutyric acid (GABA) supplementation on the endogenous auxin contents and their biosynthesis genes in the leaves of healthy and ‘*Ca*. L. asiaticus’-infected Valencia sweet orange (*C. sinensis*). (**A**–**C**) Representative chromatograms of indole-3-acetic acid (IAA), indole-3-propionic acid (IPA), and indole-3-butyric acid (IBA), respectively, from healthy and *Ca*. L. asiaticus’-infected (infected) leaves without GABA treatment (Control) or after the treatment with 10 mM GABA (treated). (**D**–**F**) Endogenous contents (ng g^−1^ FW) of IAA, IPA, and IBA, respectively, from healthy and *Ca*. L. asiaticus’-infected (infected) leaves without GABA treatment (Control) or after the treatment with 10 mM GABA (treated). The minimum and the maximum values are presented by whiskers, while thick horizontal lines specify the median. Boxes show the interquartile ranges (25th to 75th percentile of the data), white dots represent the raw data (*n* = 6), and the gray shading represents the corresponding violin plot. Presented *p*-values are based on a two-tailed *t*-test to statistically compare each pair of treatments. Statistical significance was established as *p* < 0.05. (**G**) Schematic representation of the auxins biosynthesis pathway. (**H**) Relative gene expression of auxins biosynthesis genes in healthy and *Ca*. L. asiaticus’-infected (infected) leaves without GABA treatment (Control) or after the treatment with 10 mM GABA (GABA treated). The transcript levels were organized using two-way hierarchical cluster analysis (HCA) using standardized gene expression patterns using Ward’s minimum variance method and presented as a heat map. Rows represent different genes, while columns represent different treatments. Low and high expression levels are indicated with green and red colors, respectively (see the scale at the top right corner). The full list of expressed genes, names, accession numbers, and primers is available in Appendix A. (**I**–**L**) Relative gene expression of anthranilate synthase alpha subunit 1 (*CsASA*), indole-3-acetaldehyde oxidase-like (*CsAO1*; aka Acetaldehyde oxidase), tryptophan aminotransferase-related protein 4-like (*CsTAA4*), and indole-3-pyruvate monooxygenase YUCCA8 (*CsYUC8*), respectively. Data presented means ± standard deviation (mean ± SD) of six biological replicates. Presented *p*-values are based on a two-tailed *t-*test to statistically compare each pair of treatments. Statistical significance was established as *p* < 0.05.

**Figure 4 plants-12-03647-f004:**
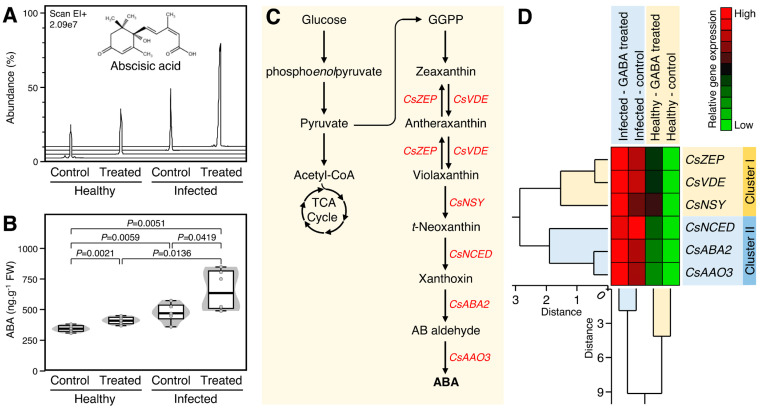
Effect of γ-aminobutyric acid (GABA) supplementation on the endogenous abscisic acid (ABA) levels and its biosynthesis genes in the leaves of healthy and ‘*Ca*. L. asiaticus’-infected Valencia sweet orange (*C. sinensis*). (**A**) Representative chromatograms of ABA from healthy and *Ca.* L. asiaticus’-infected (infected) leaves without GABA treatment (Control) or after the treatment with 10 mM GABA (treated). (**B**) Endogenous ABA contents (ng g^−1^ FW) from healthy and *Ca.* L. asiaticus’-infected (infected) leaves without GABA treatment (Control) or after the treatment with 10 mM GABA (treated). The minimum and the maximum values are presented by whiskers, while thick horizontal lines specify the median. Boxes show the interquartile ranges (25th to 75th percentile of the data), white dots represent the raw data (*n* = 6), and the gray shading represents the corresponding violin plot. Presented *p*-values are based on a two-tailed *t-*test to statistically compare each pair of treatments. Statistical significance was established as *p* < 0.05. (**C**) Schematic representation of the ABA biosynthesis pathway. (**D**) Relative gene expression of ABA biosynthesis genes in healthy and *Ca.* L. asiaticus’-infected (infected) leaves without GABA treatment (Control) or after the treatment with 10 mM GABA (GABA treated). The transcript levels were organized using two-way hierarchical cluster analysis (HCA) using standardized gene expression patterns using Ward’s minimum variance method and presented as a heat map. Rows represent different genes, while columns represent different treatments. Low and high expression levels are indicated with green and red colors, respectively (see the scale at the top right corner). The full list of expressed genes, names, accession numbers, and primers is available in Appendix A.

**Figure 5 plants-12-03647-f005:**
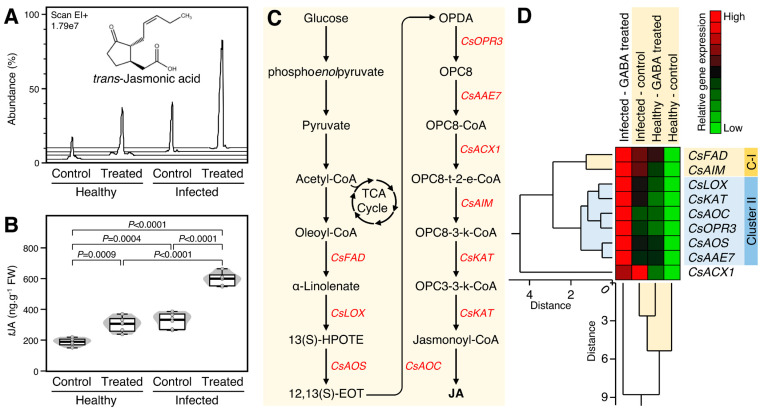
Effect of γ-aminobutyric acid (GABA) supplementation on the endogenous *trans*-jasmonic acid (*t*JA) levels and its biosynthesis genes in the leaves of healthy and ‘*Ca*. L. asiaticus’-infected Valencia sweet orange (*C. sinensis*). (**A**) Representative chromatograms of *t*JA from healthy and *Ca*. L. asiaticus’-infected (infected) leaves without GABA treatment (Control) or after the treatment with 10 mM GABA (treated). (**B**) Endogenous *t*JA contents (ng g^−1^ FW) from healthy and *Ca*. L. asiaticus’-infected (infected) leaves without GABA treatment (Control) or after the treatment with 10 mM GABA (treated). The minimum and the maximum values are presented by whiskers, while thick horizontal lines specify the median. Boxes show the interquartile ranges (25th to 75th percentile of the data), white dots represent the raw data (*n* = 6), and the gray shading represents the corresponding violin plot. Presented *p*-values are based on a two-tailed *t*-test to statistically compare each pair of treatments. Statistical significance was established as *p* < 0.05. (**C**) Schematic representation of the *t*JA biosynthesis pathway. (**D**) Relative gene expression of *t*JA biosynthesis genes in healthy and *Ca*. L. asiaticus’-infected (infected) leaves without GABA treatment (Control) or after the treatment with 10 mM GABA (GABA treated). The transcript levels were organized using two-way hierarchical cluster analysis (HCA) using standardized gene expression patterns using Ward’s minimum variance method and presented as a heat map. Rows represent different genes, while columns represent different treatments. Low and high expression levels are indicated with green and red colors, respectively (see the scale at the top right corner). The full list of expressed genes, names, accession numbers, and primers is available in Appendix A.

**Figure 6 plants-12-03647-f006:**
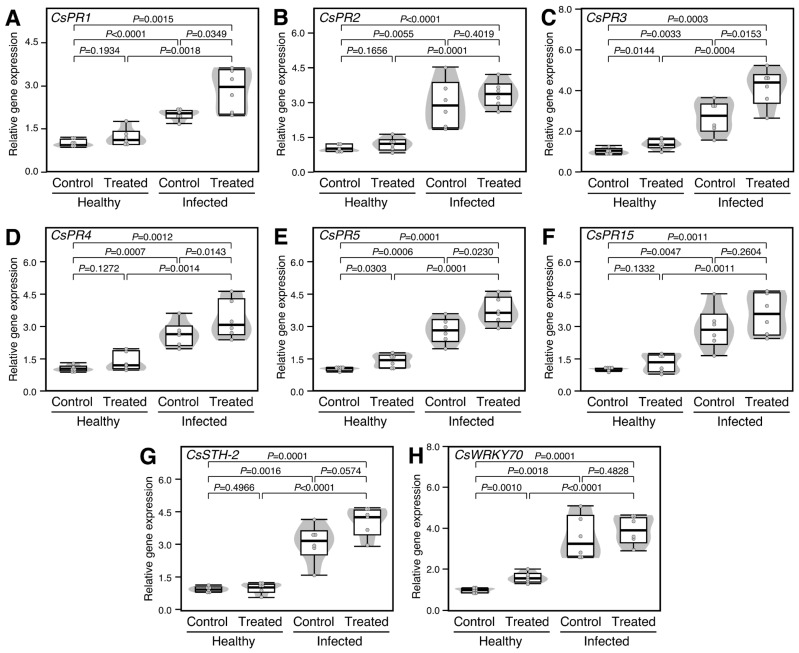
Effect of γ-aminobutyric acid (GABA) supplementation on the relative gene expression of pathogenesis-related proteins (PRs) in the leaves of healthy and ‘*Ca*. L. asiaticus’-infected Valencia sweet orange (*C. sinensis*). (**A**–**F**) Relative gene expression of six PRs included *CsPR1*, *CsPR2*, *CsPR3*, *CsPR4*, *CsPR5*, and *CsPR15*, respectively. (**G**) Relative gene expression of pathogenesis-related protein STH-2-like (*CsSTH*-2) and (**H**) Relative gene expression of WRKY70 transcription factor (*CsWRKY70*), an activator of SA-dependent defense. The minimum and the maximum values are presented by whiskers, while horizontal thick lines specify the median. Boxes show the interquartile ranges (25th to 75th percentile of the data), white dots represent the raw data (*n* = 6), and gray shading represents the corresponding violin plot. Presented *p*-values are based on a two-tailed *t*-test to statistically compare each pair of treatments. Statistical significance was established as *p* < 0.05. The full list of expressed genes, names, accession numbers, and primers is available in Appendix A.

**Figure 7 plants-12-03647-f007:**
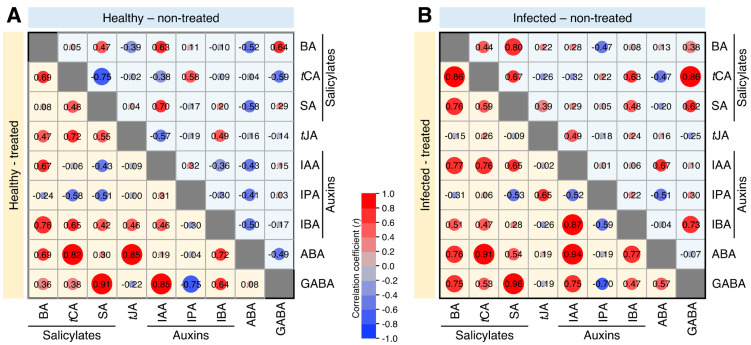
Correlation matrix among different phytohormone groups and endogenous GABA detected in Valencia sweet orange (*C. sinensis*) leaves without GABA treatment (Control) or after the treatment with 10 mM GABA (treated). (**A**) healthy leaves and (**B**) *Ca*. L. asiaticus’-infected leaves. Correlation coefficients (r) based on Pearson product-moment correlation are indicated by the color intensity and circle size. The color scale, in the middle of the graph, illustrates the correlation coefficients and their corresponding colors. BA: benzoic acid, *t*CA: *trans*-cinnamic acid, SA: salicylic acid, *t*JA: *trans*-jasmonic acid, IAA: indole-3-acetic acid, IPA: indole-3-propionic acid, IBA: indole-3-butyric acid, ABA: abscisic acid, and GABA: γ-aminobutyric acid.

**Figure 8 plants-12-03647-f008:**
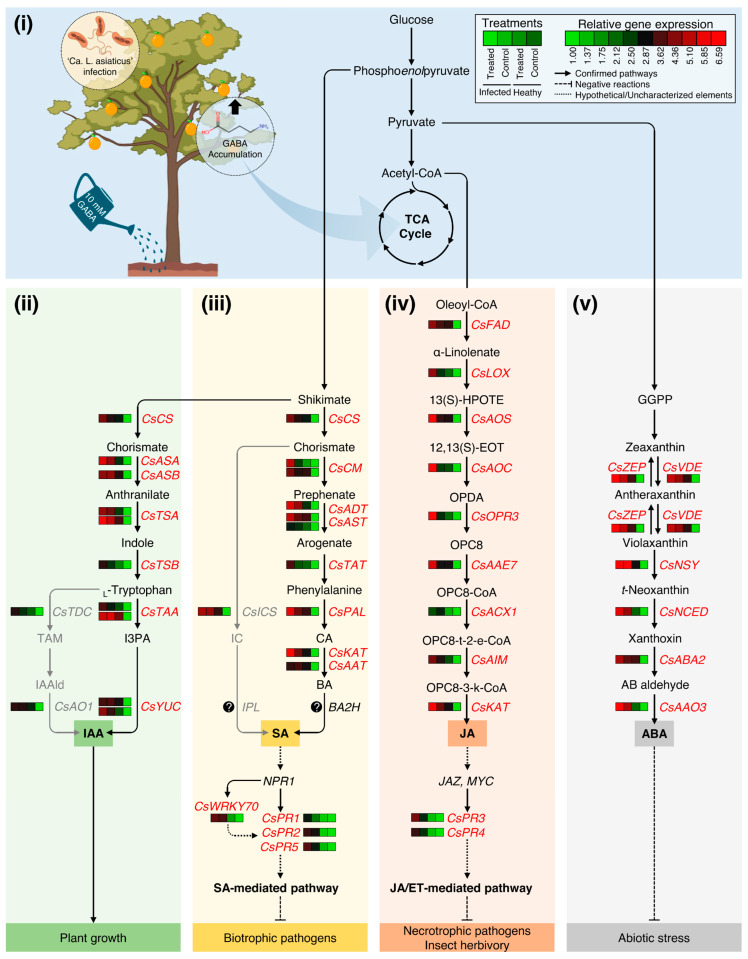
Hypothetical model for the potential effects of GABA supplementation on different phytohormonal pathways in Valencia sweet orange (*C. sinensis*) and their roles in citrus defense responses. In this model, we proposed that exogenous GABA application via root drench might enhance citrus response to the infection with the bacterial pathogen ‘*Ca*. L. asiaticus’ via modulation of different phytohormone groups, particularly stress-associated phytohormones. Briefly, (**i**) exogenous GABA application boosts the accumulation of endogenous GABA in the leaves of both healthy and ‘*Ca*. L. asiaticus’-infected plants. GABA accumulation in the leaves benefits the activation of a phytohormones-based multi-layered defensive system via promoting the endogenous content of auxins, SA, JA, ABA, and their biosynthetic genes. (**ii**) GABA accumulation stimulates the auxin biosynthesis in GABA-treated ‘*Ca*. L. asiaticus’-infected plants via the activation of the indole-3-pyruvate (I3PA) pathway, not via the tryptamine (TAM)-dependent pathway. Higher auxin levels are involved in enhancing the growth of HLB-affected tress via several biochemical and physiological modifications, yet to be identified. (**iii**) GABA accumulation is also associated with the upregulation of SA biosynthesis genes, particularly the PAL-dependent route, resulting in higher SA levels that activate NPR1 and subsequently *CsPR1*, *CsPR2*, *CsPR5*, and *CsWRKY70* prominent to activation of SA-mediated pathway against biotrophic phytopathogens. (**iv**) GABA supplementation induces JA biosynthesis. Enhanced JA levels are linked with both *CsPR3* and *CsPR4* which activate the JA-mediated pathway against insect herbivory and necrotrophic phytopathogens. Finally, (**v**) GABA application enhances the ABA biosynthesis pathway to protect citrus plants against abiotic stress. Solid lines with arrows indicate the well-established/confirmed pathways and the dashed lines with whiskers signify negative reactions, whereas round-dotted lines represent hypothetical mechanisms or uncharacterized elements. Relative gene expressions of phytohormones biosynthesis-related genes and pathogenesis-related proteins (PRs) are presented as a heatmap using the non-standardized gene expression patterns using Ward’s minimum variance method. Low and high expression levels are indicated with green and red colors, respectively (see the scale at the top right corner). The full list of expressed genes, names, accession numbers, and primers is available in Appendix A.

## Data Availability

All data supporting the findings of this study are available within the paper and/or within its Appendix A published online.

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
