# Peer review of "Gamma-Aminobutyric Acid Supplementation Boosts the Phytohormonal Profile in ‘*Candidatus* Liberibacter asiaticus’-Infected Citrus"

_plants, 2023, doi:10.3390/plants12203647_

Round 1

Reviewer 1 Report

The authors reported the influences of GABA treatment on the phytohormone changes in HLB-infected citrus trees, which have great potential to provide basis for the control of this uncurable citrus disease. The experiments were well designed and the results obtained in this study will provide basis for the HLB control. Detailed comments are as follows.

(1) The authors put forward that exogenous GABA application might be a promising alternative eco-friendly strategy that helps citrus trees battle HLB. 

However, the authors did not show the influence of GABA application on the HLB pathogen titter and the symptoms in diseased trees. Would you please add the results for the HLB bacteria titter differences between the HLB diseased trees before and after GAGA treatment?

(2) There are some typos in the manuscript. For example:

L11: 'asiaticus' should be deleted.

 L22: 'tress'--'trees'.

 L34: 'vector-borne' should be corrected to 'vector-transmitted'

 L71-72: 'GABA shunt' that should not be in italic.

 L617: 'commonly-grown commonly grown'?

The language of this manuscript is good except some typos.

Author Response

Reviewer 1: Comments and Suggestions for Authors

Journal:                       Plants (ISSN 2223-7747)

Manuscript ID:           plants-2651453

Title:                           Gamma-aminobutyric acid supplementation boosts the phytohormonal profile in ‘Candidatus Liberibacter asiaticus’-infected citrus

The authors reported the influences of GABA treatment on the phytohormone changes in HLB-infected citrus trees, which have great potential to provide basis for the control of this uncurable citrus disease. The experiments were well designed and the results obtained in this study will provide basis for the HLB control. Detailed comments are as follows.

Response: Firstly, thank you very much for your time and efforts in reviewing our manuscript. We appreciate all your comments and suggestions, which enhanced the quality of the manuscript. We have addressed all your comments in the attached file, point-by-point, with no exception. We believe all of those have been addressed in a satisfactory manner.

(1) The authors put forward that exogenous GABA application might be a promising alternative eco-friendly strategy that helps citrus trees battle HLB. However, the authors did not show the influence of GABA application on the HLB pathogen titter and the symptoms in diseased trees. Would you please add the results for the HLB bacteria titter differences between the HLB diseased trees before and after GAGA treatment?

Response: DONE, The effect of GABA application (10 mM GABA) on the bacterial population of ‘Ca. L. asiaticus’ within the detached leaves of Valencia sweet orange was investigated using quantitative PCR (qPCR) and expressed as cycle threshold (CT) values, negatively reflecting the bacterial population within the infected tissues. Briefly, although GABA supplementation did not affect the CT values in healthy leaves, it significantly increased the CT values of treated infected leaves indicating a lower bacterial population of ‘Ca. L. asiaticus’ (Kindly see Figure 1C).

(2) There are some typos in the manuscript. For example:

L11: 'asiaticus' should be deleted.

Response: DONE, since HLB was proposed to be associated with three ‘Candidatus iberibacter’ species, the word 'asiaticus' was replaced with “sp.”

 L22: 'tress'--'trees'.

Response: DONE, the word “trees” was corrected, thank you.

 L34: 'vector-borne' should be corrected to 'vector-transmitted'

Response: DONE, the word “borne” was replaced with “transmitted”

 L71-72: 'GABA shunt' that should not be in italic.

Response: DONE, thank you

 L617: 'commonly-grown commonly grown'?

Response: DONE, thank you

Reviewer 2 Report

The manuscript entitled "Gamma-aminobutyric acid supplementation boosts the phyto-hormonal profile in ‘Candidatus Liberibacter asiaticus’-infected citrus" demonstrated that exogenous GABA application can potentially be uses as an eco-friendly strategy that helps citrus trees battle HLB. The manuscript is well written and is suitable for publication.

My only concern is that the authors stated in their aims "GABA supplementation might be a sustainable, eco-friendly, and cost-effective therapeutic solution against HLB disease." The authors need to clearly state what the aims and objectives are, as well as possible outcomes.

Author Response

The manuscript entitled "Gamma-aminobutyric acid supplementation boosts the phyto-hormonal profile in ‘Candidatus Liberibacter asiaticus’-infected citrus" demonstrated that exogenous GABA application can potentially be uses as an eco-friendly strategy that helps citrus trees battle HLB. The manuscript is well written and is suitable for publication.

Response: Firstly, thank you very much for your time and efforts in reviewing our manuscript. We appreciate all your comments and suggestions, which enhanced the quality of the manuscript. We have addressed all your comments in the attached file, point-by-point, with no exception. We believe all of those have been addressed in a satisfactory manner.

My only concern is that the authors stated in their aims "GABA supplementation might be a sustainable, eco-friendly, and cost-effective therapeutic solution against HLB disease." The authors need to clearly state what the aims and objectives are, as well as possible outcomes.

Response: DONE, as recommended by “Reviewer 1’, The effect of GABA application (10 mM GABA) on the bacterial population of ‘Ca. L. asiaticus’ within the detached leaves of Valencia sweet orange was investigated using quantitative PCR (qPCR) and expressed as cycle threshold (CT) values, negatively reflecting the bacterial population within the infected tissues. Briefly, although GABA supplementation did not affect the CT values in healthy leaves, it significantly increased the CT values of treated infected leaves indicating a lower bacterial population of ‘Ca. L. asiaticus’ (Kindly see Figure 1C).

Collectively, the new added findings, along with the data presented in our manuscript, support out statement that "GABA supplementation might be a sustainable, eco-friendly, and cost-effective therapeutic solution against HLB disease.". Anyway, the “aims and objectives” of the study were rewritten to ensure clarity.

Reviewer 3 Report

Thanks for sharing your manuscript.

I am okay with accepting the manuscript in its current form. 

Due to the time conflict, I didn't have enough time to review it thoroughly, please double-check your manuscript. I have just intended to encourage the authors to perfection.

Author Response

Thanks for sharing your manuscript.

I am okay with accepting the manuscript in its current form. Due to the time conflict, I didn't have enough time to review it thoroughly, please double-check your manuscript. I have just intended to encourage the authors to perfection.

Response: Thank you very much for your time and efforts in reviewing our manuscript. We appreciate all your nice words and encouragement to perfection.